# Noise Injection Node Regularization for Robust Learning

**Noam Levi[§] & Tomer Volansky**
Department of Physics
Tel Aviv University
Tel Aviv, Israel
{noam,tomerv}@mail.tau.ac.il

**Itay M. Bloch[§]**
Berkeley Center for Theoretical Physics,
University of California and
Theoretical Physics Group,
Lawrence Berkeley National Laboratory,
Berkeley, CA, U.S.A.
itayblochm@berkeley.edu

**Marat Freytsis**
NHETC, Department of Physics and Astronomy
Rutgers University
Piscataway, NJ, U.S.A.
marat.freytsis@rutgers.edu

## Abstract

We introduce Noise Injection Node Regularization (NINR), a method of injecting structured noise into Deep Neural Networks (DNN) during the training stage, resulting in an emergent regularizing effect. We present theoretical and empirical evidence for substantial improvement in robustness against various test data perturbations for feed-forward DNNs when trained under NINR. The novelty in our approach comes from the interplay of adaptive noise injection and initialization conditions such that noise is the dominant driver of dynamics at the start of training. As it simply requires the addition of external nodes without altering the existing network structure or optimization algorithms, this method can be easily incorporated into many standard architectures. We find improved stability against a number of data perturbations, including domain shifts, with the most dramatic improvement obtained for unstructured noise, where our technique outperforms existing methods such as Dropout or $L_2$ regularization, in some cases. Further, desirable generalization properties on clean data are generally maintained.

## 1 Introduction

Nonlinear systems often display dynamical instabilities which enhance small initial perturbations and lead to cumulative behavior that deviates dramatically from a steady-state solution. Such instabilities are prevalent across physical systems, from hydrodynamic turbulence to atomic bombs (see Jeans & Darwin (1902); Parker (1958); Chandrasekhar (1961); Drazin & Reid (2004); Strogatz (2018) for just a few examples). In the context of deep learning (DL), DNNs, once optimized via stochastic gradient descent (SGD), suffer from similar instabilities as a function of their inputs. While remarkably successful in a multitude of real world tasks, DNNs are often surprisingly vulnerable to perturbations in their input data as a result (Szegedy et al., 2014). Concretely, after training, even small changes to the inputs at deployment can result in total predictive breakdown.

One may classify such perturbations with respect to the distribution from which training data is implicitly drawn. This data is typically assumed to have support over (the vicinity of) some low-dimensional submanifold of potential inputs, which is only learned approximately due to the discrete nature of the training set. To perform well during training, a network need only have well-defined behavior on the data manifold, accomplished through training on a given data distribution. However, data seen on deployment can display other differences with respect to the training set, as illustrated

---

[§]Equal contribution

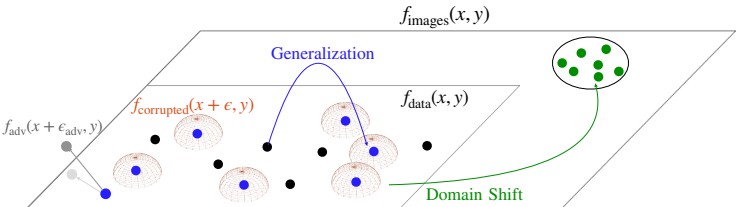

Figure 1: Illustration of perturbations to data inputs with respect to the joint probability distribution manifold of features and labels. Points indicate {sample, label} pairs $\{\boldsymbol{x}, \boldsymbol{y}\}$, where different colored points correspond to samples drawn from different marginal distributions. **Black** points represent pairs from a training dataset $\{\boldsymbol{x}_i, \boldsymbol{y}_i\}_{i=1}^{N}$, with the **red** spheres indicating corrupted inputs, determined by shifted distribution functions $f_{\text{corrupted}}(x + \epsilon, y)$. The **gray** arrow represents an adversarial attack, performed by ascending up the gradient of the network output to reach the closest decision boundary, while generalization from training to test data is depicted as interpolation from black to **blue** points. Finally, domain shift is a shift in the underlying distribution on the same manifold, depicted by the **green** arrow and points.

in Fig. 1. These distortions introduce vulnerabilities that are a crucial drawback of trained DNNs, making them susceptible to commonly occurring noise which is ubiquitous in real-world tasks. By studying how networks dynamically act to mitigate the negative effects of input noise, we identify a novel dynamical regularization method starting in a noise-dominated regime, leading to more robust behavior for a range of data perturbations. *This is the central contribution of this work.*

**Background:** Regularization involves introducing additional constraints in order to solve an ill-posed problem or to prevent over-fitting. In the context of DL problems, different regularization schemes have been proposed (for a review, see Kukačka et al. (2018) and references therein). These methods are designed to constrain the network parameters during training, thereby reducing sensitivity to irrelevant features in the input data, as well as avoiding overfitting. For instance, weight norm regularization ($L_2$, $L_1$, etc.) (Cortes & Vapnik, 1995; Zheng et al., 2003) can be used to reduce overfitting to the training data, and is often found to improve generalization performance (Hinton, 1987; Krogh & Hertz, 1991; Zhang et al., 2018). Alternatively, introducing stochasticity during training (*e.g.*, Dropout (Srivastava et al., 2014)), has become a standard addition to many DNN architectures, for similar reasons. These methods are mostly optimized to reduce the generalization error from training to test data, under the assumption that both are sampled from the same underlying distribution (Srivastava et al., 2014). Here, we propose a new method which is instead tailored for robustness. Our method relies on noise-injection, that actively reduces the sensitivity to uncorrelated input perturbations.

**Our contribution:** In this paper, we employ Noise Injection Nodes (NINs), which feed random noise through designated optimizable weights, forcing the network to adapt to layer inputs which contain no useful information. Since the amount of injected noise is a free parameter, at initialization we can set it to be anything from a minor perturbation to the dominant effect, leading to a system breakdown for extreme values. The general behavior of NINs and how they probe the network is the main goal of Levi et al. (2022), while we focus here on their regularizing properties in different noise injection regimes. The results of Levi et al. (2022) are explicitly recast in the context of regularization in App. C for a linear model, which captures the main insights.

Our study suggests that within a certain range of noise injection parameter values, this procedure can substantially improve robustness against subsequent input corruption and partially against other forms of distributional shifts, where the maximal improvement occurs for *large noise injection magnitudes* approaching the boundary of this window, above which the training accuracy degrades to random guessing. To the best of our knowledge, this regime has not been previously explored.

In the following, we analyze how the addition of NINs produces a regularization scheme which we call Noise Injection Node Regularization (NINR). The main features of NINR are enhanced stability, simplicity, and flexibility, without drastically compromising generalization performance. In order to demonstrate these features, we consider two types of feed-forward architectures: Fully Connected Networks (FCs) and Convolutional Neural Networks (CNNs), and use various datasets to train the systems. We compare NINR robustness improvement with standard regularization methods, as well as performance of these systems when using input corruption during training (CDT). Our results

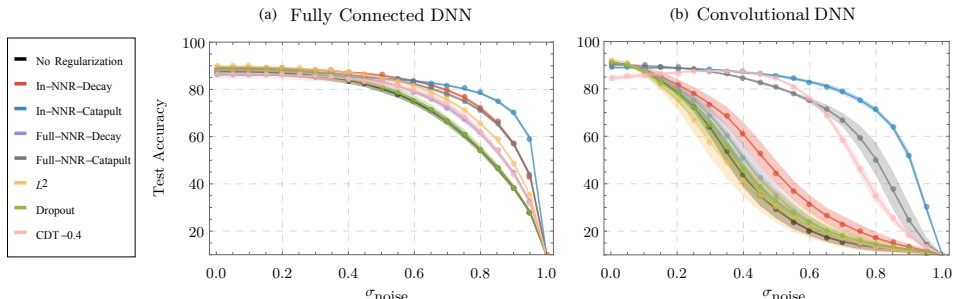

Figure 2: **Robustness against random input perturbations** tested on FC network (**left**) and CNN (**right**). Test accuracy vs. the scale of input noise corruption defined in Eq. (7) is shown for $L_2$, Dropout, in-NINR, full-NINR and CDT with $\sigma_{\text{noise}} = 0.4$. Shades indicate 2 standard deviations estimated over 10 distinct runs. For the input-NINR (full-NINR) fully-connected implementations we take $\sigma_\epsilon = 51.8 \ (16.4)$ in the decay phase and $\sigma_\epsilon = 231.6 \ (51.8)$ in the catapult phase. Similarly, for the convolutional implementations we take $\sigma_\epsilon = 2.8 \ (0.9)$ in the decay phase and $\sigma_\epsilon = 87.5 \ (62)$ in the catapult phase. This key result illustrates that NINR significantly increases the robustness of generic architectures trained on the FMNIST dataset, while marginally affecting generalization ($\sigma_{\text{noise}} = 0$). Comparing the CDT and input-NINR curves demonstrates the advantage of our regularization method. While both techniques perform similarly well on data corruption of $\sigma_{\text{noise}} = 0.4$, CDT is significantly worse on clean data. This is a result of the CDT network being forced to fit both noise and data, without the ability to suppress the latter, a crucial attribute of NINR. Here, the learning rate is fixed to $\eta = 0.05$ with mini-batch size $\mathcal{B} = 128$. Each training run is performed for 500 SGD training epochs in total, or until 98% training accuracy has been achieved. For further details, see Sec. 3 and App. A.

can easily be generalized to other architectures and more complex NINR topologies. In Fig. 2, we present our main results, comparing networks trained on the FMNIST dataset and demonstrating improved robustness against input perturbations without compromising generalization on clean data.

The paper is organized as follows. In Sec. 2 we briefly review important analytical and empirical results that are explored in depth in the work of Levi et al. (2022), demonstrating in this work how NINs implicitly generate adaptive regularization terms in the loss function. In Sec. 3, we empirically study the effectiveness of NINR. We begin by evaluating its effect on robustness against perturbations, including domain shifts and those adversarially designed, demonstrating the enhanced performance of NINR. We then verify that generalization performance on clean data is not hindered by training with NINR. We discuss related work in Sec. 4, finally concluding in Sec. 5.

## 2 NOISE INJECTION NODES REGULARIZATION

In the following sections we explain how an effective regularization scheme against input corruption naturally emerges as a consequence of adding a NIN to a DNN. First, we discuss how the NIN generates implicit regularization terms directly from computing the effective loss function. Then, we review the adaptive nature of these terms as they relate to Noise Injection Weight (NIW) dynamics during training, and discuss the expected robustness gains depending on the evolution of the NIWs.

### 2.1 EMERGENT REGULARIZATION TERMS

In order to see how NINs generate implicit regularization terms, we study a vanilla feed-forward DNN setup. Consider a supervised learning problem modeled by a neural network optimized under SGD, with an associated single sample loss function, $\mathcal{L} : \mathbb{R}^{d_{\text{in}}} \to \mathbb{R}$. The loss depends on the model parameters $\boldsymbol{\theta} = \{W^{(\ell)}, b^{(\ell)} | \ell = 0, ..., N_L - 1\}$, where $N_L$ is the number of layers, and the weights and biases associated with a given layer are $W^{(\ell)} \in \mathbb{R}^{d_\ell \times d_{\ell+1}}$, $b^{(\ell)} \in \mathbb{R}^{d_{\ell+1}}$. At each SGD iteration, a mini-batch $\mathcal{B}$ consists of a set of labeled examples, $\{(\boldsymbol{x}_i, \boldsymbol{y}_i)\}_{i=1}^{|\mathcal{B}|} \in \mathbb{R}^{d_{\text{in}}} \times \mathbb{R}^{d_{\text{label}}}$. The addition of a NIN in a given layer, $\ell_{\text{NI}}$, corresponds to a random scalar input, $\epsilon \in \mathbb{R}$, sampled repeatedly for each SGD training epoch from a chosen distribution[1], connected via NIWs $W_{\text{NI}} \in \mathbb{R}^{1 \times d_{\ell_{\text{NI}}+1}}$. We define for a given layer $\ell$, the preactivation $\boldsymbol{z}^{(\ell)} = W^{(\ell)} \boldsymbol{x}^{(\ell)} + b^{(\ell)}$, therefore the addition of a NIN to a dense layer results in a translation to the preactivation at $\ell_{\text{NI}}$, as $\boldsymbol{z}^{(\ell_{\text{NI}})} \to \boldsymbol{z}^{(\ell_{\text{NI}})} + \epsilon W_{\text{NI}}$.

---

[1]One may also generate $\epsilon$ only once, before training. We empirically find no difference between the two options, which is expected from large batch averaging. For $|\mathcal{B}| \lesssim 10$, differences begin to emerge.

The batch-averaged loss function including a NIN can be written as a series expansion[2] in the noise translation parameter $\epsilon W_{\mathrm{NI}}$,

$$L(\boldsymbol{\theta}, W_{\mathrm{NI}}) = \frac{1}{|\mathcal{B}|} \sum_{\{\boldsymbol{x}, \boldsymbol{y}, \epsilon\} \in \mathcal{B}} \mathcal{L}(\boldsymbol{\theta}, W_{\mathrm{NI}}; \boldsymbol{x}, \epsilon, \boldsymbol{y}) = \frac{1}{|\mathcal{B}|} \sum_{\{\boldsymbol{x}, \boldsymbol{y}, \epsilon\} \in \mathcal{B}} e^{\epsilon W_{\mathrm{NI}}^T \nabla_{\boldsymbol{z}^{(\ell_{\mathrm{NI}})}}} \mathcal{L}(\boldsymbol{\theta}; \boldsymbol{x}, \boldsymbol{y}). \quad (1)$$

Equation (1) follows from noting that the NIN induced translation can be written as an operator. For further details see App. B. Expanding in the parameter $\epsilon W_{\mathrm{NI}}$, we obtain an infinite series given by

$$L(\boldsymbol{\theta}, W_{\mathrm{NI}}) = L(\boldsymbol{\theta}) + \sum_{k=1}^{\infty} \mathcal{R}_k(\boldsymbol{\theta}, W_{\mathrm{NI}}). \quad (2)$$

Here, $L(\boldsymbol{\theta})$ is the loss function in the *absence* of any NIN, while $\mathcal{R}_k$ are batch-averaged derivatives of the loss function with respect to the preactivations at the noise injected layer,

$$\mathcal{R}_k(\boldsymbol{\theta}, W_{\mathrm{NI}}) \equiv \frac{1}{|\mathcal{B}|} \sum_{\{\boldsymbol{x}, \boldsymbol{y}, \epsilon\} \in \mathcal{B}} \frac{(\epsilon W_{\mathrm{NI}}^T \cdot \nabla_{\boldsymbol{z}^{(\ell_{\mathrm{NI}})}})^k}{k!} \mathcal{L}(\boldsymbol{\theta}; \boldsymbol{x}, \boldsymbol{y}). \quad (3)$$

These functions are products of the moments of the injected noise, the values of the NIWs themselves, and preactivation derivatives of the loss function in the absence of injected noise.

It is impossible to estimate when a perturbative analysis in $\epsilon$ is valid without specifying $\mathcal{L}(\boldsymbol{\theta}; \boldsymbol{x}, \boldsymbol{y})$, as all $\mathcal{R}_k$ may become equally important, or the series itself may not converge. Furthermore, since we will be interested in rather large values of $\epsilon$, where the effect of higher $\mathcal{R}_k$ terms is noticeable, the validity of the perturbative calculation is called into question even further. However, in order to gain intuition, we first study how the training procedure is altered by the NIN in the limit of small $\epsilon \ll 1$. To make further progress, we will later validate our analysis below using a combination of empirical tests, and an investigation of a linear toy model where $\mathcal{R}_k = 0$ for $k > 2$. For sufficiently small $\epsilon$ and analytic activation and loss functions, the series converges and the full loss is well-approximated by the first two leading terms in $\epsilon$. For the rest of this work, we consider noise sampled from a distribution with zero mean, relaxing this assumption only for some empirical results in App. D.2. Under this assumption the first two terms can be cast into simple forms,

$$\mathcal{R}_1 = W_{\mathrm{NI}}^T \cdot \langle \epsilon g_{\ell_{\mathrm{NI}}} \rangle, \qquad \mathcal{R}_2 = \frac{1}{2} W_{\mathrm{NI}}^T \langle \epsilon^2 \mathcal{H}_{\ell_{\mathrm{NI}}} \rangle W_{\mathrm{NI}} . \quad (4)$$

Here, batch averaging is denoted by $\langle \cdots \rangle$, while $g_{\ell_{\mathrm{NI}}} = \nabla_{\boldsymbol{z}^{(\ell_{\mathrm{NI}})}} \mathcal{L}(\boldsymbol{\theta}, \boldsymbol{x}, \boldsymbol{y})$ and $\mathcal{H}_{\ell_{\mathrm{NI}}} = \nabla_{\boldsymbol{z}^{(\ell_{\mathrm{NI}})}} \nabla_{\boldsymbol{z}^{(\ell_{\mathrm{NI}})}}^T \mathcal{L}(\boldsymbol{\theta}, \boldsymbol{x}, \boldsymbol{y})$ are the network-dependent *local* gradient and *local* Hessian, respectively. As proven in Levi et al. (2022), the magnitude of $\mathcal{R}_1$ can then be estimated using $\langle \epsilon g_{\ell_{\mathrm{NI}}} \rangle^2 \sim \sigma_\epsilon^2 \langle g_{\ell_{\mathrm{NI}}}^2 \rangle / |\mathcal{B}|$, where $\langle g_{\ell_{\mathrm{NI}}}^2 \rangle$ is the vector of the batch-averaged squared values of the local gradients and $\sigma_\epsilon^2$ is the variance of the injected noise[3], while $\mathcal{R}_2$ may be estimated using $\langle \epsilon^2 \mathcal{H}_{\ell_{\mathrm{NI}}} \rangle \approx \sigma_\epsilon^2 \langle \mathcal{H}_{\ell_{\mathrm{NI}}} \rangle$ up to corrections scaling as $O(\sqrt{1/|\mathcal{B}|})$.

While $\mathcal{R}_1$ may take both positive and negative values, the sign of $\mathcal{H}_{\ell_{\mathrm{NI}}}$ depends on the network architecture. Since the spectrum of the local Hessian is generally unknown, our analytical results are only valid for certain limiting cases. Particularly, we focus on the case of Mean Squared Error (MSE) loss and linear activation functions, where we find that the local Hessian $\mathcal{H}$ is a positive semi-definite (PSD) matrix, implying that $\mathcal{R}_2$ is a strictly non-negative penalty term, and $\mathcal{R}_k$ terms with $k > 2$ vanish identically. For the motivated case of piecewise linear activations, it was shown in Botev et al. (2017) that the local Hessian is PSD, aside from non-analytical points, hinting that $\mathcal{R}_2$ acts as a regularizer for these networks as well. This implies that for networks with piecewise linear activations and MSE loss, an analysis similar to ours below, which keeps only the first two terms in the expansion of Eq. (2), is expected to hold not only for small $\epsilon$, but also for large values. We will use this construction to understand how these terms evolve during training in the next section.

---

[2]In practice, piecewise analytic activation functions such as ReLU are often used, and if the noise causes the crossing of a non-analytic point, the above expansion receives corrections. Empirically, we find this subtlety to not change any of our qualitative conclusions.

[3]We note that the noisy loss function Eq. (1) is invariant under the simultaneous rescaling of $w_{\mathrm{NI}} \to \lambda w_{\mathrm{NI}}$ and $\epsilon \to \lambda^{-1} \epsilon$. Nonetheless, the SGD optimization equations are not invariant under this transformation, implying, in particular, that the value of the injected noise variance, $\sigma_\epsilon^2$, is a relevant parameter, not degenerate with the initialization values of the NIW. As a consequence, in order to fully explore the parameter space of noise injection, the noise (or more precisely, its variance) cannot be assumed to be small, and large noise injection values must be considered.

Figure 3: **NIW dynamics during training for the various phases** discussed in Sec. 2.2, for a single hidden layer FC network with ReLU activations trained on the full FMNIST dataset, as specified in App. A. Here, we show the evolution of the NIWs norm (**blue**) as well as the hidden layer weights norm $|W^{(\ell_{NI}+1)}|$ (**red**) against the training (**violet**) and test (**green**) loss (**solid**) and accuracy (**dashed**). **Left to right**: The NIN magnitude determines the phase of the system, ranging from the smallest amount in the decoupled phase, to an overwhelming amount in the divergent phase. The behavior displayed by the NIWs, as well as the loss corroborates the predictions discussed in Sec. 2.2 and App. C, with experimental details in App. A.

The interpretation of $\mathcal{R}_1$ and $\mathcal{R}_2$ can now be made clear: $\mathcal{R}_1$ induces a constrained random walk for in the norm of noise injection weights as well as for the data weights at layers $\ell > \ell_{NI}$, with a step size that changes according to the local gradient during training. On the other hand, $\mathcal{R}_2$, which doesn't depend on $|\mathcal{B}|$, can be understood as a straightforward regularization term for the local Hessian, working to reduce its eigenvalues. These results imply that in the limit of large batch size, and in particular full batch SGD (*i.e.*, gradient descent), regularization via $\mathcal{R}_2$ is dominant. [4].

Further understanding of why pushing the local Hessian to smaller eigenvalues is expected to reduce the sensitivity to noise corruption comes by looking at the loss for corrupted inputs. Consider therefore a network *without* a NIN but with corrupted inputs, described by the substitution, $x \to x + \delta$, with $\delta$ a random vector. To arrive at similar expressions to Eqs. (1) to (4), one can transform the preactivations $z^{(0)} \to z^{(0)} + W^{(0)}\delta$ to obtain,

$$L(\boldsymbol{\theta})|_{\boldsymbol{x} \to \boldsymbol{x} + \boldsymbol{\delta}} = \frac{1}{|\mathcal{B}|} \sum_{\{\boldsymbol{x}, \boldsymbol{y}\} \in \mathcal{B}} e^{\boldsymbol{\delta}^T W^{(0)} \cdot \nabla_{\boldsymbol{z}^{(0)}}} \mathcal{L}(\boldsymbol{\theta}; \boldsymbol{x}, \boldsymbol{y}), \tag{5}$$

Under the assumption that the components of the vector $\boldsymbol{\delta}$ are drawn i.i.d. from $\mathcal{N}(0, \sigma_\delta^2)$, the first two terms above assume simple forms[5], similar to Eq. (4),

$$\mathcal{R}_1 = \langle \boldsymbol{\delta}^T W^{(0)} \cdot g_0 \rangle, \qquad \mathcal{R}_2 = \frac{1}{2}\sigma_\delta^2 \operatorname{Tr}\left((W^{(0)})^T \langle \mathcal{H}_0 \rangle W^{(0)}\right). \tag{6}$$

As before, for sufficiently large $|\mathcal{B}|$, $\mathcal{R}_1$ is subdominant and the main regularization term due to the noise is dictated by $\mathcal{H}_0$. Thus if a NIN is inserted to the first layer, it will act to reduce $\mathcal{H}_0$ and thereby reduce the sensitivity to data corruption. Furthermore, since DNN structure in general, and loss function in particular, couples the input layer to all succeeding layers, $\mathcal{H}_0$ contains information about deeper layers and will benefit from reducing the local Hessian away from the input layer. In Sec. 3 we show results for NINs coupled to the input layer or to all layers. The above readily generalizes in this setup, resulting in multiple emergent regularization terms, which we briefly discuss in App. B.

Despite the similarities in their descriptions, we stress that a system trained on corrupted data and a system with a NIN are not the same. In the former the noise cannot be dynamically reduced without dramatically altering the optimization trajectory, implying that the DNN is not expressive enough to memorize the full data information (Ziyin et al., 2022). Conversely, in the latter, the noise has its own weights and the system can therefore improve by suppressing them without harming generalization. Nonetheless, both systems are driven towards regions with smaller local Hessian eigenvalues.

## 2.2 EVOLUTION OF NOISE INJECTION WEIGHTS

The dynamical nature of the NINR and the corresponding NIWs strongly depends on the noise distribution, parameterized in this study by $\sigma_\epsilon$. While the NIWs are updated with each learning

---

[4]In fact, it is shown in Levi et al. (2022) that all odd-terms in the expansion Eq. (2), are suppressed by the square root of the batch size, while the even terms are not.

[5]We comment on the slight subtlety of biases in Eq. (6). In any reasonable scenario, biases would not be corrupted, but if bias is treated as the zeroth component of $\boldsymbol{x}$, the zeroth component of $\boldsymbol{\delta}$ should be $\equiv 0$. Taking this into account, the trace operation of Eq. (6) should not sum over the zeroth dimension.

step, only under certain conditions is their impact on the network performance actively suppressed as the training progresses. Below we briefly describe four distinct phases of the NIWs. These are illustrated in Fig. 3, where we show the evolution of the relevant quantities (weights, loss, accuracy) for a model trained on FMNIST, demonstrating the different behavior in each phase. A complete treatment of these phases is discussed in Levi et al. (2022), while a brief derivation relating them with regularization is given in App. C for a linear network.

**Decoupled phase**. For $\sigma_\epsilon \ll 1$ one has $\mathcal{R}_1 \gg \mathcal{R}_2$, and the correction to the loss function may assume positive and negative contributions. As a consequence, the NIWs follow a small-step random walk without substantially affecting the behavior of the network.

**Decay phase**. For larger but not too large $\sigma_\epsilon$, one may ensure $\mathcal{R}_2 > \mathcal{R}_1$ at initialization while the NIN can still be treated perturbatively. In this regime, the NIWs initially experience exponential decay until $\mathcal{R}_2 \sim \mathcal{R}_1$, at which point they evolve according to the stochastic gradient. It is in this phase that one can begin to see noticeable improvement in robustness, with only minor slowing of the training. Increasing $\sigma_\epsilon$ boosts the improvement until another phase is encountered.

**Catapult phase**. The discrete nature of the training algorithm will result in a stiff numerical regime at sufficiently large $\sigma_\epsilon$. Above a critical value (for the linear network discussed in App. C we find $\sigma_{\epsilon,\text{cat}} \sim 2d_{\ell_{\text{NI}}}/\eta$ where $d_{\ell_{\text{NI}}}$ is the dimension of the NIN layer and $\eta$ is the learning rate), the effect of the NIN on the network is so significant that it causes an initial increase of the data weights, which in turn leads to an exponential increase for the loss function, followed by a recovery to a new minimum[6]. The improvement in robustness is most extreme in this phase; however, the convergence of the network is slowed somewhat, rendering the usefulness of this phase to only some applications. It is possible that a scheduled increase of the training rate after the recovery from the initial increase in the data weights could speed up the convergence. We leave such investigation to future work.

**Divergent phase**. Further increasing $\sigma_\epsilon$ leads the DNN to a breakdown of the dynamics, where the network is unable to suppress the NIN and thus cannot learn any information.

The above discussion of phases as a function of $\sigma_\epsilon$ should be taken as schematic. Other hyperparameters, such as the batch size, may also influence the phase diagram. Nonetheless, we empirically observe these phases repeating across multiple architectures and tasks, and find them to broadly capture the evolution of the NIWs. Overall, the decay and catapult phases are expected to produce an increase in robustness against input perturbations, and we empirically verify this expectation in the following sections. While we only have an analytic prediction of $\sigma_{\epsilon,\text{cat}}$ for a simple linear network, in other architectures it can also be obtained empirically using only the training data.

## 3 EXPERIMENTS

In this section, we empirically show the effect of NINR on robustness for the different phases of noise injection, following similar methodologies to Hoffman et al. (2019). After discussing the two different architectures used in this paper, we begin our investigation by demonstrating that in certain cases NINR provides a significant increase in robustness against corruption of input data by random perturbations. We then discuss the performance of NINR for domain shifts, demonstrating its effectiveness. Next, we verify that NINR does not drastically reduce the network accuracy at the original task (*e.g.*, before corruption). This is equivalent to ensuring the generalization properties of the network are not harmed due to the addition of NINs. In the main text we present results mostly for the FMINST dataset (Xiao et al., 2017). These results also extend to more complex scenarios, demonstrated in similar experiments for the CIFAR-10 (Krizhevsky et al., 2014) dataset in App. E, while evidence for improvement against adversarial attacks is given in App. D as well as results for other noise distributions and optimizers beyond SGD.

Throughout this section, we compare NINR to both unregularized DNNs, and networks explicitly regularized using $L_2$ or Dropout. We also compare NINR to implicit regularization by training with varying amounts of input data corruption. For all of our experiments, we use either an FC or a CNN (see Fig. 2 and App. A for full details). We optimize using vanilla SGD with cross-entropy loss. We preprocess the data by subtracting the mean and dividing by the variance of the training data, as is done for all subsequent datasets. The learning rate is fixed to $\eta = 0.05$ with mini-batch

---

[6]An analogous phase related to the size of the training step was discussed in Lewkowycz et al. (2020).

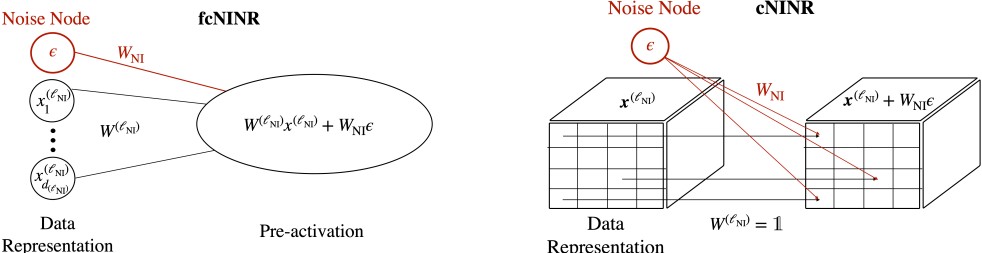

Figure 4: **Left:** An illustration of a fully connected NINR (fcNINR) for which a Noise Injection Node is appended to a representation $x^{(\ell)}$. **Right:** Implementation of NINR for a convolutional network (cNINR), where the Noise Injection Node is connected pixel-wise to the image representation $x^{(\ell)}$, to be subsequently fed into a convolutional layer.

size $\mathcal{B} = 128$. Each training run is performed for 500 SGD training epochs in total, or until 98% training accuracy has been achieved, unless otherwise specified. All test accuracy evaluations are done with the NIN output set to 0, *i.e.*, $\epsilon = 0$. The model parameters $\theta, W_{\mathrm{NI}}$ are initialized at iteration $t = 0$ using a normal distribution as $\sigma_{\theta_0}^2, \sigma_{W_{\mathrm{NI},0}}^2 = 1/d_\ell, 1/d_{\ell_{\mathrm{NI}}}$. The hyperparameters are chosen to match reference implementations: the $L_2$ regularization coefficient (weight decay) is set to $\lambda_{\mathrm{WD}} = 5 \cdot 10^{-4}$ and the dropout rate is set to $p_{\mathrm{drop}} = 0.5$. When using $L_2$ or Dropout, they are applied at/after each layer. When using input CDT as a regularization method, we corrupt the input data according to Eq. (7) below. We further stress that once a NIN has been added to the network, no further modifications to the training algorithm or architecture are required, and after choosing where to connect the NIN, the only free parameter is the injected noise variance $\sigma_\epsilon^2$.

## 3.1 REALIZATIONS IN DIFFERENT ARCHITECTURES

The way in which NINR is implemented depends on the type of layer to which the NIN is connected. Here, we comment on the two different realizations of NINR used in our experiments and depicted in Fig. 4. For both realizations we consider two distinct topologies: either we add a NIN at the input layer (in-NINR) or we couple the NIN to every hidden layer including the input one (full-NINR).

**Fully Connected Layers**   In the case of dense FC layers, we implement NINR (which we denote by fcNINR) by extending the input vector by an additional noisy pixel $\epsilon$, initialized randomly per sample at each training epoch, and densely connecting the modified input vector to the next layer. The theoretical discussion in Sec. 2 was derived for a realization of this type.

**Convolutional Layers**   Connecting a NIN at the input of a convolutional layer raises the need for a procedure for Convolutional NINR (cNINR). Since FC layers are insensitive to the input image geometry, taking $x \to \{x, \epsilon\}$ is tantamount to adding a noise mask for the entire input. In a CNN, the same interpretation can be maintained by adding the noise to the input directly in a pixel-wise fashion, $x^{(\ell)} \to x^{(\ell)} + W_{\mathrm{NI}} \cdot \epsilon$, which is subsequently fed into the convolutional layer. Importantly, this modification preserves the form of the original layer, while converging to the original $x^{(\ell)}$ for either $\sigma_\epsilon \to 0$ or $||W_{\mathrm{NI}}|| \to 0$. This can also be thought of as adding an auxiliary layer that is a non-dynamic identity matrix from the perspective of all data weights, while being densely connected from the perspective of the NIN[7].

## 3.2 ROBUSTNESS AGAINST DISTRIBUTIONAL SHIFTS

### 3.2.1 INPUT CORRUPTION

Often, training is done with examples taken in ideal conditions, which would not always exist in real-world data. This implies that the test data would be sampled from a distribution that is identical to the one trained on, albeit with an added noise component. To test the stability of networks against natural corruption, we perturb each test input image according to

$$x_i \to \sqrt{1 - \sigma_{\mathrm{noise}}^2} \, x_i + \sigma_{\mathrm{noise}} \delta_i, \tag{7}$$

---

[7]As our procedure for cNINR preserves the structure of the original $x^{(\ell)}$, it can be easily applied for other architectures beyond convolutional layers, including densely connected layers.

Table 1: **A test of domain shift**, trained on MNIST up to $100\%$ training accuracy and evaluated on the USPS test dataset, comparing $L_2$, Dropout, in-NINR, and full-NINR as discussed in the text. We exclude CDT from this table, as it is not as prevalent as the other regularization schemes, and is tailored for random noise. The fully-connected and CNN NINR noise magnitudes are those of Fig. 2. Errors indicate $2\sigma$ confidence intervals over 10 distinct runs for full training. We find the test accuracy on the full clean MNIST dataset to be similar across all regularization schemes, with $\sim 97\%$ and $\sim 98\%$ for FC and CNN respectively.

| | None | $L_2$ | Dropout | in-NINR (Decay) | in-NINR (Catapult) | full-NINR (Decay) | full-NINR (Catapult) |
|---|---|---|---|---|---|---|---|
| FC(%) | $82.3 \pm 2.0$ | $82.5 \pm 1.2$ | $\mathbf{87.5 \pm 1.2}$ | $82.1 \pm 1.6$ | $84.1 \pm 2.0$ | $82.0 \pm 2.0$ | $83.3 \pm 1.4$ |
| CNN(%) | $80.3 \pm 4.8$ | $80.5 \pm 3.8$ | $83.8 \pm 4.6$ | $82.6 \pm 4.6$ | $88.5 \pm 3.4$ | $82.3 \pm 4.0$ | $\mathbf{89.7 \pm 2.0}$ |

where each component of the perturbation vector is drawn from $\mathcal{N}(0, 1)$. In all cases except CDT, the networks are trained using clean FMNIST training data, but their accuracy is evaluated on corrupted FMNIST testing data.

In Fig. 2, we demonstrate that models trained with NINR are more robust against the noise defined above compared to those trained via other regularization methods. This verifies our expectation that in the decay phase, NINR improves stability, at least as well as CDT for large corruption, while (unlike CDT) it does not degrade generalization performance for small corruption or clean data. We also note that noise injection offers the best results for robustness within the catapult regime. However, to arrive at the same accuracy on the clean test dataset, more training epochs are generally required. Lastly, we empirically observe that in-NINR in the catapult phase offers the greatest improvement in stability against input corruption for both networks, as in-NINR most closely resembles input data corruption. We repeat this experiment for the CIFAR-10 dataset in App. E, where the similar trends persist, though at the cost of a longer training time in the catapult phase.

### 3.2.2 Domain Shift

Another test of the generalization properties induced by NINR can be realized by considering Domain Shift problems. Here, we consider the generalization between two different datasets, representing different marginal distributions, by training models with NINR on the MNIST dataset, and testing their performance on data drawn from a new target domain distribution: the USPS test set (Hull, 1994). In order to match the input dimensions of the MNIST data, we follow the original rescaling and centering done in LeCun et al. (1998). The USPS images were size normalized to fit a $20 \times 20$ pixel box while preserving their aspect ratio, and then centered in a $28 \times 28$ image field, followed by the standard preprocessing procedure.

The results are presented in Table 1. We observe generalization improvement for both FC and convolutional networks when using different regularization schemes, compared to unregularized networks. Of particular interest are the gains obtained when implementing both in- and full-NINR in the catapult phase, with the convolutional network. As the architecture becomes more complex, the improvements from NINR's adaptive scheme becomes more pronounced. The enhanced performance implies that NINR in the catapult phase could prove very beneficial for domain adaptation tasks. This is not entirely surprising as the USPS dataset is expected to lie close to the MNIST training set in distribution space, as there are no new correlated features such as several different digits in one image. Further experiments for domain shift adaptation on the MNIST-C dataset can be found in App. F. For other datasets, with large distributional shifts away from MNIST, and novel input correlations, we do not expect NINR to generically outperform other regularization methods.

### 3.2.3 Generalization to Test Data

In this section we report some effects of NINR on generalization from training to test data. While we showed above that NINR can substantially improve network performance on corrupted data, it is also important that it does not fundamentally impair the network's generalization properties.

Generically, introducing input corruption during training to increase robustness can be shown to have a negative effect on generalization on clean data. This is unsurprising as it appears that the network essentially memorizes the noise (Zhang et al., 2016), which is clearly not part of the true data distribution. As the learning process with NINR inherently leads to a suppression of the noise

Table 2: **Generalization on clean test data**, evaluated on the FMNIST dataset. Comparison is made between $L_2$, Dropout, in-NINR, full-NINR and CDT, and we highlight regularization methods which worsen generalization by italicizing. For CDT, two values ($\sigma_{\text{noise}} = 0.2$, $0.4$) for the amount of corruption are considered. The fully-connected and CNN NINR noise parameters are those used in Fig. 2. Errors indicate $2\sigma$ confidence intervals over 10 distinct runs. The NINR implementations, except perhaps the in-NINR at the catapult phase, have comparable generalization performance with to the rest of the regularization schemes, aside from CDT, where performance is diminished as the network learns the noisy distribution rather than the original one.

| | None | $L_2$ | Dropout | in-NINR (Decay) | in-NINR (Catapult) | full-NINR (Decay) | full-NINR (Catapult) | CDT (0.2) | CDT (0.4) |
|---|---|---|---|---|---|---|---|---|---|
| FC(%) | $87.7 \pm 2.6$ | $89.9 \pm 3.4$ | $88.3 \pm 0.5$ | $89.1 \pm 0.7$ | $86.2 \pm 0.8$ | $88.5 \pm 1.8$ | $88.2 \pm 0.6$ | $86.6 \pm 0.9$ | $85.5 \pm 3.8$ |
| CNN(%) | $91.0 \pm 1.0$ | $92.2 \pm 0.7$ | $91.0 \pm 1.1$ | $91.0 \pm 1.2$ | $89.0 \pm 0.6$ | $91.0 \pm 0.8$ | $90.0 \pm 0.2$ | $84.6 \pm 2.6$ | $84.1 \pm 6.4$ |

during the late stages, its generalization capabilities are expected to be far less affected. We verify this by comparing the performance of a network trained with NINR against networks trained with $L_2$, Dropout, CDT, and against unregularized DNNs.

In Table 2 we show generalization performance on the FMNIST test set for the FC and CNN architectures using the full dataset, consisting of $60\,000$ training examples with a $60/40$ training/validation split. Our main observation is that optimizing with NINR in the decay phase, as with the commonly used $L_2$ and dropout regularizers, leads to performance on clean data with in-domain test samples as least as good as the unregularized case. (In no case is the performance better at a statistically significant level.) We note that some degradation occurs when training with noise injection in the catapult phase, for a fixed number of training epochs. This degradation can be ameliorated by training for a longer period. Contrasting NINR with CDT, Table 2 clearly demonstrates that generalization is compromised for the latter, as the network cannot distinguish data from noise, learning the corrupted distribution. We further verify these results for CIFAR-10 in App. E.

## 4 RELATED WORK

Noise injection during training as a method of enhancing robustness has been proposed in various configurations in the literature. These include adding noises to input data (Hendrycks et al., 2019; Gao et al., 2020; Liu et al., 2021), activations, outputs, weights, gradients (Holmström & Koistinen, 1992; Reed & Marks, 1999; Neelakantan et al., 2015; You et al., 2019) and more. Most studies keep the amount of injected noise fixed, while we allow the network to reduce its effect during training.

Our study expands upon these works, consolidating empirical evidence with analytical insights. Our main contributions are twofold: We provide analytic expressions for the implicit regularization terms generated within our scheme, as well as estimating their effects during training. When applied to specific architectures, this allows us to predict when NINR is expected to be most effective. Additionally, we probe a novel phase of learning, starting with a large amount of noise injection and leading to a greater improvement in robustness against input corruption. Works by Rakin et al. (2018) and Xiao et al. (2021) follow similar reasoning, though both are limited, by construction, to a small amount of noise injection, and are more empirically driven. Rakin et al. (2018) and Rusak et al. (2020) also feature complex custom update steps which are less adaptable to other architectures.

## 5 CONCLUSIONS

In this paper, we motivated Noise Injection Node Regularization as a task-agnostic method to improve stability of models against perturbations to input data. Our method is simply implementable in any open source automatic differentiation system.

While we restricted this initial study to a single Noise Injection Node added to various layers, with a fixed scale of noise injection during training, this restriction can be relaxed, leading to potential improvements to NINR. For instance, changing the amount of injected noise during training, similar to learning rate scheduling, could aid in convergence speed while still obtaining the advantages of a large amount of noise injection.

## 6    ACKNOWLEDGEMENTS

We thank Yasaman Bahri, Kyle Cranmer, Guy Gur-Ari, and Sho Yaida for useful discussions and comments. NL would like to thank the Milner Foundation for the award of a Milner Fellowship. MF is supported by the DOE under grant DE-SC0010008 and the NSF under grant PHY1316222. MF would like to thank Tel Aviv University, the Aspen Center for Physics (supported by the U.S. National Science Foundation grant PHY-1607611), and the Galileo Galilei Institute for their hospitality while this work was in progress. The work of TV is supported by the Israel Science Foundation (grant No. 1862/21), by the Binational Science Foundation (grant No. 2020220) and by the European Research Council (ERC) under the EU Horizon 2020 Programme (ERC-CoG-2015 - Proposal n. 682676 LDMThExp).

## REPRODUCIBILITY STATEMENT

In Sec. 2, we state our theoretical results, ensuring that we state our assumptions and the limitations of the approximations we make at every step. In several instances, we rely on proofs given in other works, as well as supplement our analyses in App. C; The models and tools used for analysis in our experiments are provided in the following anonymous link: `https://anonymous.4open.science/r/NoiseInjectionNodeCode-2A68`, while explicit details regarding our experimental setup as well as a complete description of the data processing steps for the datasets we used, are given in Sec. 3 and App. A.

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

## A   NETWORK ARCHITECTURE DETAILS

Here we describe the experimental settings specific to each of the figures in the paper. All the models have been trained with cross-entropy loss unless otherwise specified.

Fig. 2(a). Fully connected, three hidden layers with width $d_{1,2,3} = 1024$, weight initialization $W^{(\ell)} \sim \mathcal{N}(0, 1/d_\ell)$, $b = 0$. ReLU activation, trained using SGD (no momentum) on FMNIST, with a learning rate of $\eta = 0.05$ and batch size $|\mathcal{B}| = 128$.

Fig. 2(b). CNN composed of 2 convolutional blocks, followed by a dense ReLU layer with width $d_3 = 2048$ and a dense connection to the prediction layer, weight initialization $W^{(\ell)} \sim \mathcal{N}(0, 1/d_\ell)$, $b = 0$. Each convolutional block is taken as Conv2D(2,2) $\rightarrow$ ELU $\rightarrow$ Batch-Norm $\rightarrow$ MaxPool(2,2), trained using SGD (no momentum) on FMNIST, with a learning rate of $\eta = 0.05$ and $|\mathcal{B}| = 128$.

Fig. 3. Fully connected, one hidden layer with $d_1 = 1024$, weight initialization $W^{(\ell)} \sim \mathcal{N}(0, 1/d_\ell)$, $b = 0$. ReLU activation, trained using SGD (no momentum) on FMNIST. with learning rate $\eta = 0.01$ and $|\mathcal{B}| = 1000$. From left to right, the injected noise is $\sigma_\epsilon^2 = \{0, 0.1 \cdot d_{\text{in}}/\eta, d_{\text{in}}/\eta, 1.8 \cdot d_{\text{in}}/\eta\}$, corresponding to the decoupled, decay, catapult, and divergent phases, respectively. Here, $d_{\text{in}} = 785$ is the dimension of the input data including a single NIN.

## B   ADDITIONAL THEORETICAL DETAILS

### B.1   DERIVATION OF THE NOISE TRANSLATED LOSS FUNCTION

Here we provide additional details on the theoretical analysis of the noise translated loss function, leading to Eq. (1). The introduction of a NIN at a specific layer $\ell_{\text{NI}}$ translates the preactivation as $z^{(\ell_{\text{NI}})} + W_{\text{NI}}\epsilon$. A single sample loss function describing the translated preactivation can be written as

$$\mathcal{L}(\boldsymbol{\theta}, W_{\text{NI}}; \boldsymbol{x}, \epsilon, \boldsymbol{y}) = \mathcal{L}(\boldsymbol{\theta}; z^{(\ell_{\text{NI}})} + W_{\text{NI}}\epsilon, \boldsymbol{y}). \tag{8}$$

Using the definition of the translation operator

$$f(x + a) = e^{a\nabla} f(x), \tag{9}$$

we explicitly compute the batch averaged loss function

$$L(\boldsymbol{\theta}, W_{\text{NI}}) = \frac{1}{|\mathcal{B}|} \sum_{\{\boldsymbol{x},\epsilon,\boldsymbol{y}\} \in \mathcal{B}} \mathcal{L}(\boldsymbol{\theta}, W_{\text{NI}}; \boldsymbol{x}, \epsilon, \boldsymbol{y}) = \frac{1}{|\mathcal{B}|} \sum_{\{\boldsymbol{x},\epsilon,\boldsymbol{y}\} \in \mathcal{B}} e^{\epsilon W_{\text{NI}}^T \nabla_{\boldsymbol{z}^{(\ell_{\text{NI}})}}} \mathcal{L}(\boldsymbol{\theta}; \boldsymbol{x}, \boldsymbol{y})$$

$$= L(\boldsymbol{\theta}) + \frac{1}{|\mathcal{B}|} \sum_{\substack{\{\boldsymbol{x},\epsilon,\boldsymbol{y}\} \\ \in \mathcal{B}}} \sum_{k=1}^{\infty} \frac{1}{k!} (\epsilon W_{\text{NI}}^T \cdot \nabla_{\boldsymbol{z}^{(\ell_{\text{NI}})}})^k \mathcal{L}(\boldsymbol{\theta}; \boldsymbol{x}, \epsilon, \boldsymbol{y}).$$

Expanding in powers of $\epsilon W_{\text{NI}}$, we obtain an infinite series given by

$$L(\boldsymbol{\theta}, W_{\text{NI}}) = L(\boldsymbol{\theta}) + \frac{1}{|\mathcal{B}|} \sum_{\{\boldsymbol{x},\epsilon,\boldsymbol{y}\} \in \mathcal{B}} \sum_{k=1}^{\infty} \frac{1}{k!} (\epsilon W_{\text{NI}}^T \cdot \nabla_{\boldsymbol{z}^{(\ell_{\text{NI}})}})^k \mathcal{L}(\boldsymbol{\theta}; \boldsymbol{x}, \epsilon, \boldsymbol{y}), \tag{10}$$

where we identify the $k \geq 1$ terms in the expansion with the implicit regularization terms defined in Eq. (3).

### B.2   NIN AT ALL LAYERS

Here we extend our theoretical derivations from the case of a NIN connected to a single layer, to a single NIN connected to all layers (the full-NINR case).

Similar to Eq. (1), we may write down the loss using the translation operator,

$$L(\boldsymbol{\theta}, W_{\text{NI}}) = \frac{1}{|\mathcal{B}|} \sum_{\{\boldsymbol{x},\boldsymbol{y},\epsilon\} \in \mathcal{B}} \left( \prod_{\ell=0}^{N_L-1} e^{\epsilon(W_{\text{NI}}^{(\ell)})^T \nabla_{\boldsymbol{z}^{(\ell)}}} \right) \mathcal{L}(\boldsymbol{\theta}; \boldsymbol{x}, \boldsymbol{y}), \tag{11}$$

where as one may expect, we now have $N_L$ vectors $W_{\text{NI}}^{(0)}, ..., W_{\text{NI}}^{(N_L-1)}$, of respective dimensions $\mathbb{R}^{1 \times d_1}, ..., \mathbb{R}^{1 \times d_{N_L}}$. Focusing once more on the leading terms in $\epsilon$, we can see that the first order

regularization term is simply

$$\mathcal{R}_1 = \sum_{\ell=0}^{N_L-1} \left(W_{\mathrm{NI}}^{(\ell)}\right)^T \cdot \langle \epsilon g_\ell \rangle, \tag{12}$$

which is the sum of the regularization terms at each layer. The second order regularization is slightly more complex, and may be written as

$$\mathcal{R}_2 = \frac{1}{2} \sum_{\ell_1=0}^{N_L-1} \sum_{\ell_2=0}^{N_L-1} (W_{\mathrm{NI}}^{(\ell_1)})^T \langle \epsilon^2 \mathcal{H}_{\ell_1 \ell_2} \rangle W_{\mathrm{NI}}^{(\ell_2)}, \tag{13}$$

with $\mathcal{H}_{\ell_1 \ell_2} \equiv \nabla_{\boldsymbol{z}^{(\ell_1)}} \nabla_{\boldsymbol{z}^{(\ell_2)}}^T \mathcal{L}(\boldsymbol{\theta}, \boldsymbol{x}, \boldsymbol{y})$.

According to Botev et al. (2017); Martens & Grosse (2015), the terms which mix different layers in the Hessian are expected to be small, thus leading us to a sum over the single-layer $\mathcal{R}_2$s of the each $\ell$. Therefore, we may use the same arguments used in the main text to estimate the scaling of the two terms with $\sigma_\epsilon$ and $|\mathcal{B}|$. Much like the single-layer case, we therefore expect $\mathcal{R}_1 \propto \sigma_\epsilon/\sqrt{|\mathcal{B}|}$ and $\mathcal{R}_2 \propto \sigma_\epsilon^2$.

Demonstrating the emergence of the phases discussed in the main text (*i.e.* the decay phase, the catapult phase, etc.) on a deep linear network for this full-NINR case is beyond the scope of this work. However, we do note that empirically they are found to be present much like in the single-layer NIN case.

## C    NINR IN A LINEAR TOY MODEL

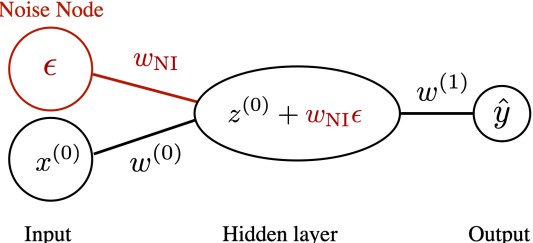

Figure 5: Illustration of the univariate linear DNN with a single input scalar, noise node, and one hidden layer.

In order to elucidate the interpretation of noise injection nodes as an emergent regularization scheme, combined with a form of constrained random walk, we employ an (over)simplified univariate linear model which captures the main features present in realistic networks. Consider a linear network (*i.e.*, linear activation functions), with a single hidden layer and no biases ($b = 0$), aiming to perform a linear regression task. The data consists of a set of training samples $\left\{(x_a, y_a) \in \mathbb{R}^2\right\}_{a=1}^m$, taken by drawing the inputs from a normal distribution of $x_a \in X \sim \mathcal{N}(0, \sigma_x^2)$, where $\sigma_x > 0$. The corresponding outputs are then given by a linear transformation $y_a = M \cdot x_a$ with $M \in \mathbb{R}$.

The noise node is added at the input, and its weight is $w_{\mathrm{NI}}$, with the input's data weight being $w^{(0)}$. The hidden layer is directly connected to the output, and has a single weight associated with it, $w^{(1)}$, as illustrated in Fig. 5. We use Mean Squared Error (MSE) loss, and for simplicity take a full-batch gradient descent, and thus our loss function is

$$L_{\mathrm{MSE}} = \frac{1}{2|\mathcal{B}|} \sum_{a \in \mathcal{B}} \left(w^{(1)}(w^{(0)} \cdot x_a + w_{\mathrm{NI}}\epsilon_a) - y_a\right)^2. \tag{14}$$

Performing an explicit averaging, we can further simplify to

$$L_{\mathrm{MSE}} \simeq \frac{1}{2} \left[ 2w^{(1)} w_{\mathrm{NI}} \left(w^{(1)} w^{(0)} - M\right) \sigma_x \sigma_\epsilon \frac{\Phi}{\sqrt{|\mathcal{B}|}} + (w^{(1)} w^{(0)} - M)^2 \sigma_x^2 + (w^{(1)})^2 w_{\mathrm{NI}}^2 \sigma_\epsilon^2 \right], \tag{15}$$

where $\Phi$ is a random variable with zero mean and unit variance[8]. From Eq. (15), we can easily see that the optimal solution is achieved for the weights $w_*^{(1)}w_*^{(0)} = M$, $w_{\mathrm{NI},*} = 0$. We can clearly read $\mathcal{R}_{1,2}$ off Eq. (4) by separating them from the unperturbed loss function

$$L_{\mathrm{MSE}}(w_{\mathrm{NI}}, \boldsymbol{\theta}) \simeq L_{\mathrm{MSE}}(\boldsymbol{\theta}) + \mathcal{R}_1(w_{\mathrm{NI}}, \boldsymbol{\theta}) + \mathcal{R}_2(w_{\mathrm{NI}}, \boldsymbol{\theta}), \tag{16}$$

while $\mathcal{R}_k$ vanish for $k > 2$. The various terms in Eq. (16) are given by

$$L_{\mathrm{MSE}}(\boldsymbol{\theta}) = \frac{1}{2}(w^{(1)}w^{(0)} - M)^2\sigma_x^2,$$

$$\mathcal{R}_1(w_{\mathrm{NI}}, \boldsymbol{\theta}) = w_{\mathrm{NI}}\langle \epsilon g_{\ell_{\mathrm{NI}}}\rangle = w_{\mathrm{NI}}w^{(1)}\left(w^{(1)}w^{(0)} - M\right)\frac{\sigma_x\sigma_\epsilon\Phi}{\sqrt{|\mathcal{B}|}}, \tag{17}$$

$$\mathcal{R}_2(w_{\mathrm{NI}}, \boldsymbol{\theta}) = \frac{1}{2}\sigma_\epsilon^2 w_{\mathrm{NI}}^2 \mathcal{H}_{\ell_{\mathrm{NI}}} = \frac{1}{2}(w^{(1)})^2 w_{\mathrm{NI}}^2 \sigma_\epsilon^2,$$

where we have identified the local gradient term which generates a constrained random walk for $w_{\mathrm{NI}}$, which decreases as the network approaches its data driven minimum. We also note that in the limit of an infinite batch size $|\mathcal{B}| \to \infty$ it vanishes, leaving only an effective regularization term for the hidden layer weight, namely $L_2 = \lambda(w^{(1)})^2$ where the Lagrange multiplier $\lambda = \frac{1}{2}w_{\mathrm{NI}}^2\sigma_\epsilon^2$ decreases with time as the noise weight $w_{\mathrm{NI}}$ is pushed to 0.

We may glean further insights from this linear example by studying its training dynamics for small and large noise variances. Assuming full batch gradient descent in the infinite sample limit, we neglect the local gradient contribution and focus on the coupled equations for the hidden layer weight and the noise weight, given respectively by

$$w_{t+1}^{(1)} = w_t^{(1)}(1 - \eta\sigma_\epsilon^2 w_{\mathrm{NI},t}^2) - \eta(w_t^{(1)}w_t^{(0)} - M)w_t^{(0)}\sigma_x^2,$$
$$w_{\mathrm{NI},t+1} = w_{\mathrm{NI},t}(1 - \eta\sigma_\epsilon^2 (w_t^{(1)})^2) \tag{18}$$

Assuming $M \neq 0$, without the loss of generality, we may set $M = 1$ as the equations remain invariant under reparameterization[9]. In the limit of $\sigma_\epsilon^2 \ll 1/\eta w_{\mathrm{NI},0}^2$, the equations decouple, with the data weight following the standard GD equation without noise, i.e., $w_{t+1}^{(1)} = w_t^{(1)} - \eta(w_t^{(1)}w_t^{(0)} - 1)w_t^{(0)}\sigma_x^2$, while the noise weight decays exponentially as long as $0 < |w_t^{(1)}|$ and $(w_0^{(1)})^2 > \eta\sigma_\epsilon^2/2$. Clearly, the smaller $\sigma_\epsilon$ is, the smaller the regularizing effect of the noise on the local Hessian, given by the square of the hidden layer weights in this simple model.

We expect that in this regime, the limit of continuous time GD should reproduce the correct dynamics as $\eta\sigma_\epsilon^2 \to 0$, yielding a differential equation for the noise weight

$$\dot{w}_{\mathrm{NI}}(t) = -\sigma_\epsilon^2(w^{(1)}(t))^2 w_{NI}(t), \tag{19}$$

where $\dot{x} = dx/dt$ is the continuous time derivative. The noise weight can therefore only decay.

Conversely, taking the large noise variance limit we find that the dynamics are ignorant of the original learning objective, as the resulting equations become simply coupled

$$w_{t+1}^{(1)} = w_t^{(1)}(1 - \eta\sigma_\epsilon^2 w_{\mathrm{NI},t}^2),$$
$$w_{\mathrm{NI},t+1} = w_{\mathrm{NI},t}\left(1 - \eta\sigma_\epsilon^2(w_t^{(1)})^2\right). \tag{20}$$

These equations describe a NN, trained using completely random data with no labels or learning objective, with an effective loss given by the last term in Eq. (15). In this case, we expect the continuous time limit to fail as a complete description of the possible dynamics, as $\eta\sigma_\epsilon^2$ may be large. We may demonstrate this failure by taking the continuous time limit, obtaining

$$\dot{w}_{\mathrm{NI}}(t) = -\sigma_\epsilon^2(w^{(1)}(t))^2 w_{NI}(t), \qquad \dot{w}^{(1)}(t) = -\sigma_\epsilon^2(w_{\mathrm{NI}}(t))^2 w^{(1)}(t), \tag{21}$$

implying both weights decrease in magnitude. This means the network, even for arbitrarily large $\sigma_\epsilon$ will not diverge. However, this is clearly not the case for the discrete Eq. (20), which may become stiff for sufficiently large noise variance. This numerical artifact entirely changes the weight behavior, opening up the possibility for the system to either diverge, or catapult, as discussed in Levi et al. (2022). To summarize, this simple example provides a useful test case for our main analytical derivations appearing in the main text, displaying all the expected features of NINR in a fully calculable setting.

---

[8]Additional $\mathcal{O}(\sigma_\epsilon^2/\sqrt{|\mathcal{B}|})$ corrections coming from stochastic variations in the $\sigma_\epsilon^2$ term emerge from batch-averaging but are neglected.

[9]Taking $w^{(1)} \to Mw^{(1)}$, $w_{\mathrm{NI}} \to Mw_{\mathrm{NI}}$ and $\sigma_\epsilon \to \sigma_\epsilon/M$ leaves the equations invariant.

# D    ADDITIONAL EXPERIMENTS

Throughout this section, we train the FC and the CNN using the same specifications as given in Fig. 2, unless otherwise specified. Training is performed for the minimum between 500 epochs, and the time it takes the network to reach $98\%$ training accuracy. This is done with the goal of demonstrating that NINR using a large amount of noise injection requires a longer period of training, otherwise suffering from degraded generalization performance, as discussed in the main text.

## D.1    ADVERSARIAL ATTACKS

In addition to input perturbations caused by deployment issues, natural degradation, and unexpected noise sources, targeted perturbations, meant to maximally impair the performance of a network while changing the data as little as possible, form a conceptually different concern. Quantifying what corresponds to a minimal distortion of the data is a domain-specific and somewhat subjective task. Nevertheless, standard approaches exist. One of the simplest known implementations for an adversarial attack is the white-box untargeted Fast Gradient Sign Method (FGSM) (Goodfellow et al., 2014), which transforms inputs according to

$$x \to x + \delta_{\mathrm{FGSM}} \times \mathrm{sign}(\nabla_x \mathcal{L}(\boldsymbol{\theta}; \boldsymbol{x}, \boldsymbol{y})), \tag{22}$$

where $\delta_{\mathrm{FGSM}}$ is a small positive parameter that controls the size of the perturbation. We also consider the Projected Gradient Descent (PGD) attack (Kurakin et al., 2016; Madry et al., 2017), which iterates the FGSM attack $k$ times, compounding its effect.

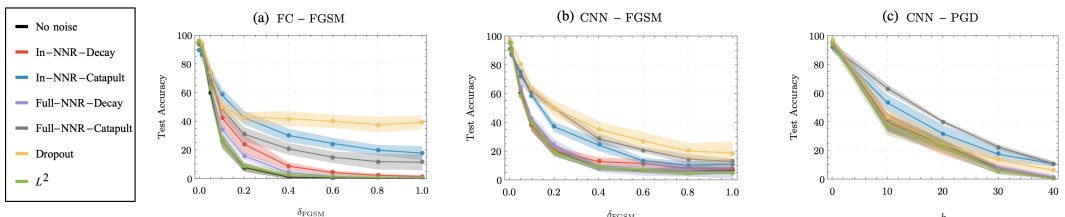

Figure 6: Robustness against adversarial attacks. **Left:** FC network, **Right:** CNN (detailed specifications are in App. A). This key result illustrates that NINR significantly increases the robustness of generic model architectures trained on the FMNIST dataset. Shades indicate 2 standard deviations estimated over 10 distinct runs.

In Fig. 6 we compare the performance of standard regularization schemes with NINR against FGSM and PGD type adversarial attacks. We find that NINR displays superior performance over $L_2$ and un-regularized nets. For FGSM attacks dropout performs best among the options tested, while for PGD attacks NINR outperforms.

These preliminary results suggest potential improvement against certain types of adversarial attacks when NINR is used. Further analysis is required to determine whether combining NINR with other regularization schemes, or changing the noise distribution during training could potentially produce a more successful scheme.

## D.2    DIFFERENT NOISE DISTRIBUTIONS

Here, we examine the effects of sampling the NINs from different noise distributions on the performance of NINR. For each different noise distribution, we repeat the tests used to produce Fig. 2, demonstrating robustness against corrupted inputs. We compare results using a uniform distribution $\epsilon \sim U(-\sigma_\epsilon, \sigma_\epsilon)$ and an asymmetric (double Gaussian peaked at $\pm\sigma_\epsilon$) distribution, for fcNINR and cNINR using DNNs trained on the FMNIST dataset.

In Fig. 7, we see that varying the noise distribution has a minimal effect on NINR as a regularization scheme, aside from the asymmetric distribution for the catapult phase. We attribute this behavior to an extreme choice of noise injection scale, where a much longer training time is required to obtain good performance for NINR.

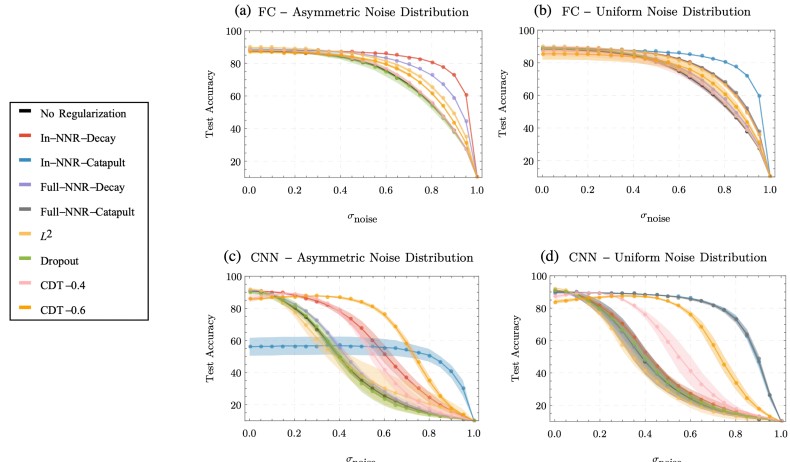

Figure 7: Robustness against random input perturbations for FC (**top row**) and convolutional (**bottom row**) networks using NINR training with different noise distributions (detailed specifications are in App. A). **Left:** Asymmetric double gaussian distribution peaked at $\pm\sigma_\epsilon$, **Right:** Uniform distribution $\epsilon \sim U(-\sigma_\epsilon, \sigma_\epsilon)$. The fully-connected and CNN NINR noise magnitudes are those of Fig. 2. Shades indicate 2 standard deviations estimated over 5 distinct runs.

## D.3 DIFFERENT OPTIMIZERS

Here, we examine the effects of changing the optimization algorithm, beyond SGD, on the performance of NINR. For each different optimizer, we repeat the tests used to produce Fig. 2, demonstrating robustness against corrupted inputs. We compare results using RMSprop (Hinton, 2012) and Adam (Kingma & Ba, 2014), for fcNINR and cNINR using DNNs trained on the FMNIST dataset. Here, we use different parameters for the different architectures and optimizers. Namely, RMSprop - $\rho = 0.9$, $\epsilon = 10^{-7}$ and $\eta = 0.0001$ for FC and $\eta = 0.001$ for CNN. Adam - $\beta_1 = 0.9$, $\beta_2 = 0.999$, $\epsilon = 10^{-7}$ and $\eta = 0.01$ for both FC and CNN, with noise injection magnitudes given in App. D.3.

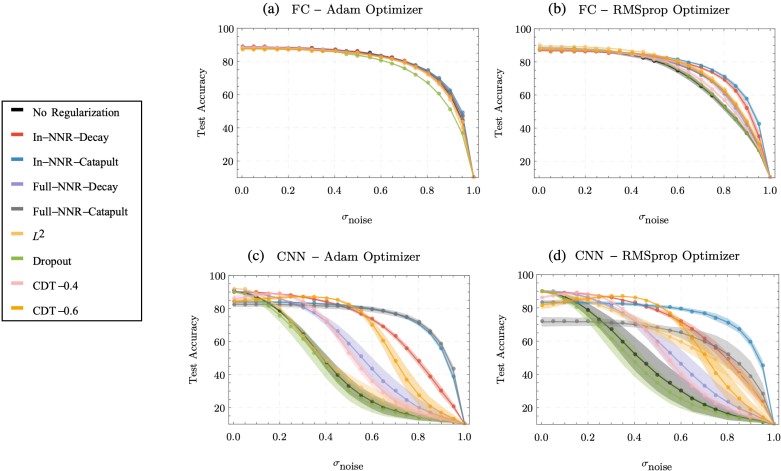

Figure 8: Robustness against random input perturbations for FCs (**top row**) and CNNs (**bottom row**) using NINR training under different optimization schemes (detailed specifications are in App. A). **Left:** Adam, trained with $\eta = 0.01$ for both FC and CNN, **Right:** RMSprop, trained with $\eta = 0.0001$ for FC and $\eta = 0.001$ for CNN. The fully-connected and CNN NINR noise magnitudes are those of App. D.3. Shades indicate 2 standard deviations estimated over 5 distinct runs.

Table 3: Amount of noise injection ($\sigma_\epsilon$) for the different architectures using RMSprop and Adam

| | | in-NINR - decay (catapult) | full-NINR - decay (catapult) |
|---|---|---|---|
| RMSprop | FC | 1158.2 (5179.9) | 366.3 (1158.2) |
| | CNN | 19.6 (619.1) | 6.19 (437.8) |
| | | in-NINR - decay (catapult) | full-NINR - decay (catapult) |
| Adam | FC | 115.8 (518) | 36.6 (115.8) |
| | CNN | 6.2 (195.8) | 1.95 (138.4) |

# E    RESULTS FOR CIFAR-10

Here, we implement cNINR, working with a CNN based on VGG style blocks described in Simonyan & Zisserman (2014). The CIFAR-10 dataset consists of color images of objects divided into 10 categories, with $32 \times 32$ pixels in 3 color channels, each pixel intensity in the range [0, 1], partitioned into $50\,000$ training and $10\,000$ test samples, which are then preprocessed similarly to the FMNIST dataset.

The network used to test NINR performance is constructed by connecting the following blocks[10]:

- Conv2D(32,3,3) → ReLU → Batch Norm → Conv2D(32,3,3) → ReLU → Batch Norm → MaxPool(2,2) → Dropout($p_{\text{drop}} = 0.2$).

- Conv2D(64,3,3) → ReLU → Batch Norm → Conv2D(64,3,3) → ReLU → Batch Norm → MaxPool(2,2) → Dropout($p_{\text{drop}} = 0.3$).

- Conv2D(128,3,3) → ReLU → Batch Norm → Conv2D(128,3,3) → ReLU → Batch Norm → MaxPool(2,2) → Dropout($p_{\text{drop}} = 0.4$).

- Dense ReLU Layer(500) → Linear Layer(10).

Optimization is done using SGD without momentum with the learning rate fixed to $\eta = 0.05$ and mini-batch size $\mathcal{B} = 128$. Each training run is performed for 500 SGD training epochs in total, or until $98\,\%$ training accuracy has been achieved.

We provide preliminary results for robustness against input-data corruption in Fig. 9. In contrast to the previous sections, the CNN used to train on CIFAR-10 contains Dropout and $L_2$ as part of its architecture, making comparison between NINR and the two redundant. Therefore, we show results for the same network with and without NINR, as well as CDT with different input corruption scales. The success of NINR is retained for in-NINR in the decay phase, while the catapult phase requires longer than 500 epochs to obtain similar generalization properties.

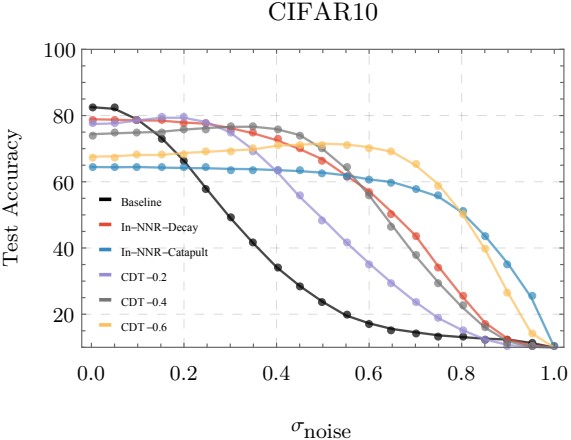

Figure 9: Robustness against random input perturbations for the network described in App. E using NINR training on CIFAR10. Here, we use the in-NINR CNN implementation, taking $\sigma_\epsilon = 17.5$ in the decay phase and $\sigma_\epsilon = 55.4$ in the catapult phase.

---

[10]Each convolutional layer admits $L_2$ weight decay regularization ($\lambda_{\text{WD}} = 10^{-4}$).

# F    RESULTS FOR MNIST-C

Here, we show some additional results for the same architectures and NINR parameters used in Sec. 3, trained on the MNIST data-set and tested on several classes of images from MNIST-C. The MNIST-C dataset (Mu & Gilmer, 2019) consists of 15 types of corruption applied to the MNIST test set, for benchmarking out-of-distribution robustness in computer vision. For testing purposes, the data is preprocessed similarly to the FMNIST dataset.

Table 4: **Domain shift performance on the MNIST-C test data**, for networks trained on the MNIST dataset. Comparison is made between $L_2$, Dropout, in-NINR, full-NINR and CDT. For CDT, two values ($\sigma_{\text{noise}} = 0.2,\ 0.4$) for the amount of corruption are considered. The fully-connected and CNN NINR noise parameters are those used in Fig. 2. The NINR implementations improve performance for data transformations which are most closely related to gaussian noise injection, as can be expected.

Fog transformation

| | None | $L_2$ | Dropout | in-NINR (Decay) | in-NINR (Catapult) | full-NINR (Decay) | full-NINR (Catapult) | CDT (0.2) | CDT (0.4) |
|---|---|---|---|---|---|---|---|---|---|
| FC(%) | 53.2 | 52.3 | 61.9 | 59.4 | 63.8 | 57.2 | 62.8 | 52.9 | 57.3 |
| CNN(%) | 71.8 | 67.3 | - | 61.8 | 62.5 | 62.9 | 57.8 | 57.8 | 62.7 |

Brightness transformation

| | None | $L_2$ | Dropout | in-NINR (Decay) | in-NINR (Catapult) | full-NINR (Decay) | full-NINR (Catapult) | CDT (0.2) | CDT (0.4) |
|---|---|---|---|---|---|---|---|---|---|
| FC(%) | 97.7 | 97.6 | 98.4 | 98.0 | 97.4 | 97.9 | 98.1 | 97.7 | 97.2 |
| CNN(%) | 98.9 | 98.7 | - | 98.7 | 98.8 | 98.8 | 98.6 | 98.1 | 97.5 |

Glass Blur transformation

| | None | $L_2$ | Dropout | in-NINR (Decay) | in-NINR (Catapult) | full-NINR (Decay) | full-NINR (Catapult) | CDT (0.2) | CDT (0.4) |
|---|---|---|---|---|---|---|---|---|---|
| FC(%) | 93.7 | 93.2 | 96.0 | 94.7 | 93.4 | 94.4 | 94.3 | 94.6 | 94.3 |
| CNN(%) | 65.1 | 54.6 | - | 60.2 | 95.2 | 61.3 | 95.6 | 80.6 | 90.2 |

Impulse Noise transformation

| | None | $L_2$ | Dropout | in-NINR (Decay) | in-NINR (Catapult) | full-NINR (Decay) | full-NINR (Catapult) | CDT (0.2) | CDT (0.4) |
|---|---|---|---|---|---|---|---|---|---|
| FC(%) | 84.3 | 84.3 | 94.6 | 93.6 | 94.1 | 89.7 | 96.1 | 86.7 | 89.7 |
| CNN(%) | 51.1 | 28.8 | - | 62.8 | 97.8 | 57.1 | 96.5 | 75.3 | 86.8 |

The results shown in Table 4 indicate improved performance when the type of image corruption applied to the MNIST images most closely resembles the injected noise. It can therefore be intuitively understood why the most dramatic performance enhancement is found for the *Impulse Noise* corruption transformation, while other corruption transformation may not benefit much from NINR. We stress that NINR can be readily modified to deal with different types of corruption by changing the

noise injection distribution, as well as incorporated with other regularization methods to compound their robustness enhancing effects.

## G  CONSTANT NOISE INJECTION

Here, we reproduce the results shown in Fig. 2, including an additional curve representing a constant input noise injection. We implement this experiment by applying dNINR at the input layer (connecting to the first hidden layer) using the large NIN variance value used for the the "catapult" phase, but keeping the NIWs static, fixed to their initialization values.

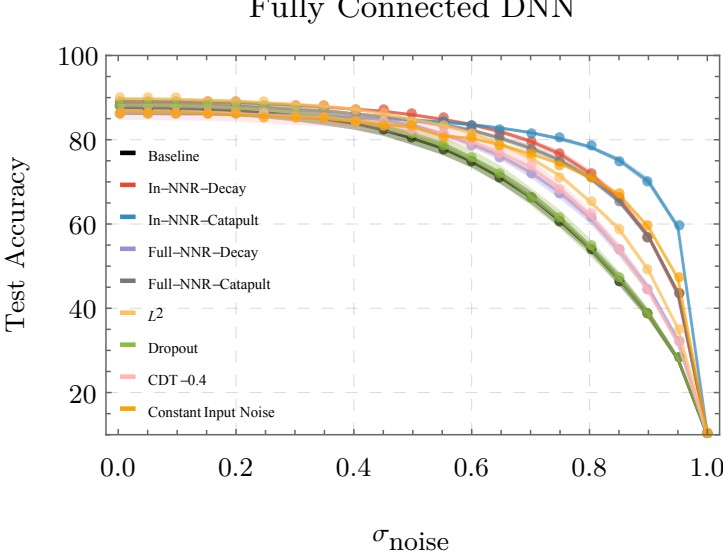

Figure 10: Robustness against random input perturbations with the same parameters used in Fig. 2. The additional orange curve represents In-NINR with $\sigma_\epsilon = 231.6$ but with fixed NIW values, which is essentially constant noise injection to the pre-activation at the first hidden layer.

