# OpenReview forum: "Noise Injection Node Regularization for Robust Learning"
_ICLR.cc/2023/Conference — ICLR 2023 poster_

### Official Review · Reviewer_KZHy · 2022-10-24

**Confidence:** 4
**Clarity, Quality, Novelty And Reproducibility:** See the above.
**Correctness:** 2
**Technical Novelty And Significance:** 2
**Empirical Novelty And Significance:** 2
**Recommendation:** 5

**Strength And Weaknesses:**

The major problem of this manuscript is its writing, which cites a lot of analytical results from Anonymous (2022). Unfortunately, I cannot find this reference from the web at all. This makes the current version of  this paper lack of self-containess. Thus, frankly, I cannot judge the novelty of this work.

**Summary Of The Paper:**

This works studies the noise injection node approach and how it improves the noise-resistance to various types of data perturbations.

**Summary Of The Review:**

Due to the issue of self-containess, I cannot give the acceptance to this paper.

---

> ### Author Response · Authors · 2022-11-19
> **Thank you for your review! We have updated the paper to provide more context and address you concern**
>
> Thank you for raising the issues of self-containedness to consider our work in proper context. You reference the main difficulty in judging our manuscript as a lack of access to Anonymous (2022) online. We completely agree with this assessment, which was certainly true at the time of the original submission and an unfortunate consequence of a dual double-blind review process. The cited paper was under double blind review and we were instructed to cite it as we did. The paper is now available on the arXiv as Ref. [1]. We have revised the citations in the re-submitted manuscript, which will hopefully clarify most of our claims and address the your primary objection.
>
> The reviewer’s further comments have also made us realize that we should have emphasized that the main features and results of the NIN dynamics, leading to the behavior discussed for the different regularization terms were already discussed in detail in App. C at the time of submission. Although only presented for a linear model, the same observations persist across all architectures and datasets
> we considered. To clarify this issue, as well as point the reader to the relevant results in the appendix, we have added a clearer statement in the second paragraph of p. 2, stating, “The general behavior of NINs and how they probe the network is the main goal of Levi et al. (2022), while we focus here on their regularizing properties in different noise injection regimes. The results of Levi et al. (2022) are explicitly recast in the context of regularization in App. C for a linear model, which captures the main insights.”
>
> We hope with these changes the self-contained nature of the the text, including appendices, is clearer, while the broader context is more readily accessible, and that upon review of the revised version you find our work suitable to acceptance.
>
> [1] [Levi et al. (2022), Noise Injection as a Probe of Deep Learning [arXiv:2210.13599]](https://arxiv.org/abs/2210.13599).

---

### Official Review · Reviewer_q36r · 2022-10-25

**Confidence:** 4
**Correctness:** 3
**Technical Novelty And Significance:** 3
**Empirical Novelty And Significance:** 2
**Recommendation:** 6

**Clarity, Quality, Novelty And Reproducibility:**

As I mentioned in the [Strength And Weaknesses], the idea does not sound novelty and there are several papers that have already proposed a similar idea. The authors have provided the code, so the experiments may be reproduced.

**Strength And Weaknesses:**

Strength:
The paper covers an interesting topic as robust representation learning is a challenging but practical problem, especially for robust image classification. The overall presentation sounds good.

Weaknesses:

1) I doubt the novelty of this paper. Injecting noises to the input or to the learned representation is not a new thing. There are already several papers covering similar topics, such as Ref1 and Ref2.

2) All experiments are carried out on very simple and easy datasets such as MNIST and FMNIST. Thus, it is not quite sure how reliable these results are and whether it is a safe way to draw conclusions from them.

3) Also for the comparisons, the authors only compare models with Dropout and L2 regularization. However, as the paper claims to learn a robust representation, the authors should consider comparing the proposed method with other robust models and other more complicated datasets such as MNIST-C and ImageNet-C for example.

4) Even on the simple dataset, the proposed method cannot consistently outperform models trained with Dropout. Thus, I do worry about the benefits of this proposed method can bring us.

5) There are some typos in the writing.


Ref1:  Liu, Aishan, et al. "Training robust deep neural networks via adversarial noise propagation." IEEE Transactions on Image Processing 30 (2021): 5769-5781.

Ref2: Rusak, Evgenia, et al. "A simple way to make neural networks robust against diverse image corruptions." European Conference on Computer Vision. Springer, Cham, 2020.



**Summary Of The Paper:**

This paper focuses on robust representation learning and attempts to propose a method that is robust to input perturbations. To achieve this goal, the authors propose Noise Injection Node Regularization (NINR) which severs as a “regularizer”. Experimental results demonstrate in some cases, this proposed method outperforms models trained with Dropout or L2 regularization.

**Summary Of The Review:**

Based on what I have mentioned in the [Strength And Weaknesses] part, I have concerns about the novelty of this paper and also its experimental evaluation does not sound comprehensive.

---

> ### Author Response · Authors · 2022-11-20
> **Thank you for your challenging review! We have updated the paper to address your concerns and provide further discussion below.**
>
> Please note that we are **breaking this response into two parts** due to the 5000 character limit and the amount of criticism to be addressed.
>
> Thank you for your probing comments, which have forced us to consider carefully how we structured our results. We are glad that you agree that the underlying question is a problem that is both challenging and of practical importance. We have made a number of changes to which we hope address or clarify some of the issues you’ve raised, and which we hope would would lead you to consider admission. We address these in more detail now, starting what what we view as the most serious.
>
> 1. **The novelty of this paper:** You correctly state and noise injection is in no way a novel concept in the regularization literature. It was never our intention to claim that noise injection or noise injection as regularization were themselves new to our results. The innovations that our approach relative to Ref1, Ref2, and the prior work we cite in the main text are:
>     * *Adaptive noise injection*. Most noise injection methods (including the references given by the reviewer) rely on corrupting inputs or representations by a constant amount of noise/adversarial corruption. Under a constant noise injection, the system must but optimize for a predetermined mixture of both noise and data, (typically) resulting in degradation of the generalization error on clean data, which is a crucial point. We explain this in detail in Eqs. (5), (6), and surrounding text. On the other hand, NINR is adaptive precisely since the NIWs are free to dynamically evolve, allowing the network to compensate for the noise by adapting weights in subsequent layers, while reducing the dependence on the precise amount of noise seen by any neuron during training by allowing the network to adjust noise weight magnitudes dynamically. The adaptive behavior not only improves the robustness without degrading generalization, but it also allows one to make use of the various phases in noise injection systems in a simple manner, without varying many hyper-parameters.
>     * *Initializing in the large-noise regime to access a previously unknown phase of learning.* To our knowledge, no other references explore noise injection to generate the “catapult” dynamics that we analyze, and for which we observe the greatest robustness gains (as is analytically motivated). This is crucially tied with the point above, since dynamically existing the catapult phase is only possible due to the adaptive mechanism above This is clearly a novel feature, going beyond the standard conclusions in the literature, which alone warrants the interest of the community in our opinion.
>     * *No required modifications of optimization method.* Generically, noise injection methods beyond the simplest approach of corrupting inputs/representations usually require nontrivial modifications of the optimization method. This fact means that incorporating these methods into existing architecture is more complex, and does not necessarily lead to the desired results on clean data.
>
>     We would also like to address your additional references to clarify our distinction from them. In Ref2, the described procedure and results are an almost identical implementation of what we call “Corruption during Training” (CDT), where indeed the input data is corrupted by some fixed amount at initialization, allowing the network training to proceed as usual. While this enhances robustness, the tradeoff is that the network learns the corrupted data, causing the deterioration of the generalization error on clean test data. We address this exact point around Eqs. (5), (6), while also showing that our method outperforms this approach in Fig. 2 (the pink curve versus the blue curve).
>
>     The approach of Ref1 is more closely related to our own, as they choose to corrupt representations, instead of inputs. Nonetheless, their method is not the same. While they do update the measure of corruption during training, this requires an adversarial training step and not simple SGD, and as a result is both more complexity than our method and has faces the same issues of training stability faced by other adversarial min-max methods, which require driving the risk to a saddle point. Additionally, they do not probe the regime of large noise, which is precisely where the most gains are expected and obtained.
>
>     We thank the reviewer for bringing these specific papers to our attention as they clearly form relevant background. We now cite Ref1 when discussing related work adding corruption to input data in Sec. 4 and Ref2 when discussing other adaptive methods and we have made revisions throughout the text in an effort to better highlight the novelty of our approach with respect to other methods.

---

> > ### Author Response · Authors · 2022-11-20
> > **Part II of our response**
> >
> > 2. **All experiments are carried out on simple datasets:** We fully understand your concerns regarding the simplicity of our datasets. We now make it clearer that while we provide empirical results on these simpler data sets in the main text, we also provide comparable results for CIFAR-10 in App. E (Fig. 9), albeit at the cost of a longer training period relative to unregulated networks for large noise variance values. We also explore the effects of different optimizers and noise distributions in the appendices, showing that the experimental results themselves are quite robust. We now make reference to these additional experiments more explicitly. Namely, p. 6, final paragraph now mentions that “these results also extend to more complex scenarios, demonstrated in similar experiments for the CIFAR-10 (Krizhevsky et al., 2014) dataset in App. E, while evidence for improvement against adversarial attacks is given in App. D as well as results for other noise distributions and optimizers beyond SGD.” and p. 8, second paragraph now includes “We repeat a similar experiment in App. E for the CIFAR10 dataset, where the similar trends persist, though at the cost of a longer training time in the catapult phase.
> >
> > 3. **Comparisons with other robust models and more complicated datasets:** You are right to point out that we primarily compared against dropout and L2, though we also compared out method against a simple noise injection method which introduces constant corruption during training. The reason to mainly limit ourselves to these comparison against these basic methods is that they do not require any alteration of the optimization algorithm, much like our own NINR method are are thus an equitable comparison of methods with low deployment cost. Other, more complicated regularization schemes are generally based on modifying both the optimization algorithm as well as the loss function/other components of the network, which is not required in NINR. Moreover, due to its simplicity, our method can be incorporated alongside any other regularization scheme, while some other methods cannot. Thank you for suggesting that we would benefit from a test against a more comprehensive suite of corruption possibilities. To this end, we now include several experiments reporting performance on MNIST-C in a new App. F, which we believe more fully demonstrates the robustness of our algorithm.
> >
> > 4. **Consistent performance compared to Dropout:** The reviewer’s point is entirely valid, however, though our method does not outperform other regularization schemes in all possible scenarios, this is a first work on this particular subject, and we believe future improvements could greatly improve performance (though even now, as can be seen, the network outperforms other methods
> > in quite a few cases). We do note that in the case of input corruption, which is what NINRs are most suited to address, the catapult phase of NINR almost universally outperforms all other methods for large enough corruptions, often by a wide margin. We have made revisions to the text highlighting these points, while also pointing out that NINR is easily combined with more standard methods, unlike more complicated adaptive noise schemes.
> >
> > 5. **Typos in the writing:** We have fixed all located typos and fixed a several cases of non-standard hyphenation. We hope at this point that all such typographical issues have been addressed.
> >
> > **Overall summary:** The reviewer has concerns regarding the novelty of the paper, as well as whether our experimental results are comprehensive enough. We have made a number of revisions to emphasize that our method differs from others by merit of its adaptive nature, its simplicity, and its ability to access novel learning dynamics by initializing learning in a noise-dominated phase, motivated
> > by theoretical considerations. Following your constructive review, we have also expanded our results with a number of additional experiments to demonstrate the gains in robustness across a wider range of settings. Finally, we are now able to cite an unblinded version of [Levi et al. (2022) (Noise Injection as a Probe of Deep Learning [arXiv:2210.13599])](https://arxiv.org/abs/2210.13599), which provides more background for our analytical analysis.

---

> > > ### Comment · Reviewer_q36r · 2022-12-05
> > > **Thanks for the clarifications**
> > >
> > > Thank you so much for the clarifications and the revision of the paper. After reading the revision, the discussions, and the clarifications, most of my previous concerns have been addressed to some degree. Thus I would like to raise my rating accordingly.

---

### Official Review · Reviewer_27TF · 2022-10-25

**Confidence:** 3
**Correctness:** 4
**Technical Novelty And Significance:** 2
**Empirical Novelty And Significance:** 3
**Recommendation:** 6

**Clarity, Quality, Novelty And Reproducibility:**

**Clarity**
---

While most of the paper is well written parts of the paper could be better written. Specifically,

1) NINs are introduced implicitly rather than explicitly. That is, the paper talks about their expansion before defining what an NIN is.

2) The paper mentions that when $\mathcal{R}_1$ is big this induces a random walk. But it is not clear what is meant by this. I think it means a random walk for the norm of the scaling parameters.

3) The paper in the introduction (first paragraph) mentions chaotic dynamical systems and represent SGD as one such chaotic system. However, the paper then proposes adding further stochasticity to the training to improve robustness. These two statements seem contradictory as the first would suggest, that this additional stochasticity would increase the variance of the model. However, the rest of paper never touches upon this.

**Quality and Novelty**
---

While the theoretical work and experiments seems reasonable, the novelty is difficult to judge because of the missing context for the work.

**Reproducible**
---

The paper seems to highly reproducible.


**Strength And Weaknesses:**

**Strengths**
---

1) I like the study on the norm of the scaling during training versus the noise variance this provides nice insights into how to set the noise. The theoretical underpinning for this comes from the Taylor expansion of the loss function with respect to the scaling parameters. The paper looks at the terms with the first derivative and the second derivative and in some simple case (such as 2 layer linear networks) derive heuristics for estimating these terms. Using the heuristics, the paper divides the variance into various regimes depending on whether the First derivative term or the second derivative dominates.

2) The proposed method seems to drastically increase robustness of Convolutional neural networks.

3) The idea of having adaptive noise is interesting. I will list this as a strength (because I don't fully know the literature) but two papers cited by the paper (Rakin et al. (2018) and Xiao et al. (2021)) might have already proposed something similar.

**Weakness**
---

I think the major weakness of the work is putting itself in context of the existing literature. Specifically, the works on adding noise to the input data being equivalent to regularization is known (Bishop 1995). Further, based on a google search there is a lot of work analyzing constant noise injection.  For example, Camuto, Willetts, Simsekli, Roberts, Holmes 2020; Poole, Dickstein, Ganguli 2014; and  Cohen, Resnfeld, Kolter 2019. While the authors do not need to cite these particular work, but more discussion on work related to constant level noise injection would be appreciated.

Further, in the experimental evaluation the authors do not compare against constant level noise injection. Since the main novelty of the paper seems to be the adaptability of the noise level, I feel that the constant noise level is a crucial baseline.

**Summary Of The Paper:**

This paper looks at a new method for adding gaussian noise to nodes during training such that the amount of noise added is a learned parameter. The paper, the does some theoretical analysis, to obtain some heuristics about the effect of the noise. This followed up by empirical work looking at the evolution of the norms of noise scaling parameter during training for different noise variances. Finally, the paper tests the robust of such networks to input data corruption.

**Summary Of The Review:**

In summary, I like the idea, the analysis and the experiment, but I am concerned about the novelty and the presentation.

---

> ### Author Response · Authors · 2022-11-19
> **Thank you for your detailed review! We have made updates and added clarifications to address your concerns**
>
> Thank you for your valuable comments. We would like to point out that the theoretical background can now be better understood, as Anonymous (2020) has been unblinded and is now cited properly in the text as [Levi et al. (2022) (Noise Injection as a Probe of Deep Learning [arXiv:2210.13599])](https://arxiv.org/abs/2210.13599). We would like to address the weaknesses and questions of clarity you have brought up.
>
> * **The context of existing literature:** We thank the reviewer for pointing us towards these specific papers dealing with constant noise injection. Clearly, we did not adequately explain that the process we referred to as CDT is equivalent to the ones discusse  in most of these works (up to minor variations). We should have also explained that due to the way in which we implemented NINR, a constant noise injection to either inputs or representations is equivalent to keeping the noise injection weights fixed, as already encompassed within our framework. The relationship between regularization and fixed training data curruption was not intended to seem novel itself, and we have made revisions to the text to clarify this point. To serve as another baseline, and demonstrate that fixed noise injections in the first layer is equivalent to CDT we have added results with fixed noise injection weights for an FC network in App. G.
> * **Implicit introduction of NINs:** We have revised the order on the last paragraph of p. 3 to make the logical flow more natural.
> * **Nature of random walk induced by $\mathcal{R}_1$:** Thank you for pointing out the unclear nature of this discussion. We have now expanded our discussion in the first paragraph of p. 5, where we now write: “The interpretation of $\mathcal{R}_1$ and $\mathcal{R}_2$ can now be made clear: $\mathcal{R}_1$ induces a constrained random walk in the norm of the noise injection weights,”
> * **Relation of chaotic dynamics and stochasticity in the introduction:** We apologize for not framing our comments on chaotic dynamics in the introduction clearly enough, leading to this apparent contradiction in the first paragraph. The chaotic dynamics are evidenced not by SGD itself, but by the behavior of functions learned under SGD. With no regularization, these functions often display instabilities with respect to small perturbations in the inputs. By adding adaptive stochasticity to the training process as we describe, the resulting learned functions become more robust and less chaotic in the sense above. To delineate this distinction more sharply, we now state in the first paragraph of the introduction that we are discussing instabilities in DNNs “as a function of their inputs.”
> * **Adaptive noise in existing literature:** While the use of adaptive noise is noted as a strength by the reviewer (and we agree!), we do wish to explicitly address the existing literature cited in the comment. We contrast our results with both of the papers mentioned in Sec. 4. Our critical insight, and where we observe the most significant improvement in robustness, is starting training in the large-noise regime, which can only be exited if the noise is adaptive. To realize that this choice of initialization is of value requires our theoretical analysis, which is lacking in both of these papers. Moreover, [Rakin et al. (2018)](https://arxiv.org/abs/1811.09310) develop a more complicated custom update step for the noise parameters in their approach, which is less adaptable to other architectures, more involved to implement, and makes training in the large-noise regime impossible by construction. Especially when facing large corruption, as can be seen in, e.g., Fig. 2(b), the catapult phase can lead to an improvement of the accuracy by a factor of few, compared to ∼10% improvements in the decay phase.
>
> We greatly appreciate the reviewer’s very useful comments, and request that they go over the revised version where their concerns have been addressed, hopefully leading to the acceptance of the paper.

---

> > ### Comment · Reviewer_27TF · 2022-11-29
> > **Thank you for the clarifications.**
> >
> > The discussion here and the response to Reviewer q36r was helpful and has helped me better understand the context and the novelty of the paper.
> >
> > Specifically, I think the contributions of the paper are looking at this high initial noise regime, and experimentally showing that this improves robustness. This does seem interesting and new.
> >
> > With the improved writing of the paper, I am now leaning towards accept.

---

### Official Review · Reviewer_J5gg · 2022-10-27

**Confidence:** 3
**Correctness:** 3
**Technical Novelty And Significance:** 2
**Empirical Novelty And Significance:** 3
**Recommendation:** 6

**Clarity, Quality, Novelty And Reproducibility:**

Additionally, I have some questions and suggestions:

* What do “uncorrelated input perturbation” and “certain window of convergence” in Page 2 mean?
* Equations (1) and (2) are the key to the rest of the analysis, but as a reader I am not sure I follow them easily. I would find a detailed derivation here or Appendix very helpful. Also, using W_{NI} weight as a vector (versus W^(l) as a matrix) is somewhat confusing.
* In Equation (6), \sigma^2_delta can be very small. How does it factor into R_2 when you say “dynamics is controlled by H_0”?
* How would the authors make sense of the performance gap between NINR applied to FC and CNN? E.g., a big jump in performance for full-NINR Catapult.

**Strength And Weaknesses:**

Neural network’s robustness is one of the most important topics in deep learning, so searching for a new regularization method that makes neural networks more robust to various kinds of scenarios such as distribution shift, adversarial attacks is definitely worthwhile. The paper presents an intuitively simple yet interesting idea motivated by mathematical and empirical insights. The experimental results are also extensive.

Regarding the weaknesses, I have some comments:

1. The novelty of this work compared to Anonymous, (2022), which I do not have the full context into.
2. It is unclear from the experimental results whether this regularization approach is more useful than other well adapted ones. For example, in Table 1, Dropout is much better than NINR-based regularization for FC. In Table 2, L2 outperforms the rest for both FC and CNN.
3. I may be mistaken, but it is unclear which layer of the neural network one should apply the NIN regularization. If applied on multiple layers, would the same analysis in Section 2.1 follow and how?




**Summary Of The Paper:**

This paper introduces a regularization method for neural networks, namely Noise Injection node Regularization (NINR). The high-level idea is to inject random noise into the network’s training at a certain layer via a learnable weight. The authors provide analyses both theoretically and empirically to show how NINR could improve the robustness.



**Summary Of The Review:**

I think this work has some merits, but I also have some concerns as given above.

---

> ### Author Response · Authors · 2022-11-20
> **Thank you for your thorough review! We have updated the paper to address your concerns.**
>
> Thank you for your careful reading and thoughtful comments. We have made several modifications to the text to address the weaknesses pointed out, which we discuss below.
>
> * **The novelty of this work compared to Anonymous (2022):** We agree and would like to apologize for the confusion. Since both works were concurrently submitted to double-blind reviews, we tried to follow the instructions of conference staff given to us to keep both anonymous in cross-referencing. The cited work is now available online and cited accordingly in the text as [Levi et al. (2022) (Noise Injection as a Probe of Deep Learning)](https://arxiv.org/abs/2210.13599). It investigates the behavior of NINs with a focus on their phase structure but without regard to their effect on robustness. Therefore, the two works have separate and distinct novelty. To contextualize this, we write on p. 2, second paragraph: “The results of Levi et al. (2022) are explicitly recast in the context of regularization in App. C for a linear model, which captures the main insights.”
> * **Is this regularization approach more useful than other well-adapted ones?:** Thank you for allowing us to clarify this point. As can be seen in Fig. 2, Our method works best when used against random corruption. In Table 1, we test performance under domain shift. It is hard to predict which methods will work best for shifted datasets in the absence of a precise definition of domain shift, but we wanted to present evidence that for more complicated architectures, the adaptive methods of NINR start to work better, which we now mention explicitly in the last paragraph on p. 8. The purpose of Table 2 is to demonstrate that unlike constant noise injection (CDT), generalization on clean test data is not degraded by NINR. It has been widely observed that across architectures that $L_2$ does best for simple generalization with no noise. To more clearly highlight the purpose of the latter comparison in the text, at the end of Sec. 3.2.3 we now explain that decay NINR and the standard methods lead to “performance on clean data with in-domain test samples as least as good as the unregularized case. (In no case is the performance better at a statistically significant level.)” Thus, this situation is comparable for better-established methods, e.g., Dropout, only there the heuristics are better developed.
> * **To which layer of the neural network one should apply NINR:** We have added a discussion in an appendix in an effort to heuristically describe the expected effects, which we reference on p. 5, second paragraph: “The above derivation readily generalizes in this setup, resulting in multiple emergent regularization terms, which we briefly discuss in App. B.” Roughly speaking, the theoretical analysis of other variations would involve a straightforward extension of Sec. 2.1 with the appearance of multiple regularization terms relevant to smaller subsections of the network involving other local Hessians.
> * **Meaning of “uncorrelated input perturbation” and “certain window of convergence”:** We have modified the text to be clearer in this respect. “Uncorrelated” is now “random.” We only meant to emphasize perturbations statistically uncorrelated with the inputs. Instead of refering to a “certain window”, the relevant text on p. 2, second paragraph now reads: “Our study suggests that within a certain range of noise injection parameter values , this procedure can substantially improve robustness against subsequent input corruption...”.
> * **Derivation of Eqs. (1) and (2). Notation for noise weight vector $W_{NI}$:** We have added a short appendix (B.1) explaining the derivation in more detail, to which we refer below Eq. (1). Regarding the vector vs. matrix definition of $W_{NI}$, in our case the dimensionality of the NIN input is 1, and $W_{NI}$ can be considered a 1 × $d_{\ell_{NI}}$ matrix or a vector. In the general case of a vector of NINs, $W_{NI}$ would be a $d_{NIN}$ × $d_{\ell_{NI}}$ matrix.
> * **Effect of $\sigma_\delta^2$ on relative size of $R_1$ and $R_2$:** Our intention was to contrast NINR and input corruption. While $R_1$ may be important, it is suppressed by $1/\sqrt{|B|}$, so for sufficiently large batch sizes, the leading regularization term is $R_2$, which is controlled by $H_0$, even for small $\sigma_\delta^2$. The text below Eq. (6), now states this explicitly.
> * **NINR performance gap between FC and CNN:** Thank you for allowing us to revisit this question. As briefly mentioned above, we do not currently have a strong statement to make regarding this jump. This situation is rather similar other regularization schemes where a predictive understanding of their performance for various tasks and architectures is still absent. However, in those cases more detailed heuristics are currently available. Planned followup work will focus on understanding NIN dynamics in more complex scenarios beyond the simple dense case. Hopefully this will shed some light on this issue.

---

### Decision · Program_Chairs · 2023-01-20

**Decision:**

Accept: poster

**Justification For Why Not Higher Score:**

The work should have been better positioned wrt recent and less recent related work at submission time.

**Justification For Why Not Lower Score:**

The authors adequately addressed the reviewers concerns.

**Metareview: Summary, Strengths And Weaknesses:**

The authors consider noise injection as a regularization method in neural networks. The authors make, both, methodological and theoretical contributions. Experimental evidence shows that the proposed approach benefits CNNs among others. The authors addressed the concerns of the reviewers during the rebuttal phase. However, I would like to emphasize that the authors to should 1/ more clearly position their work compared to related work and 2/ clarify novelty as they did in the rebuttal given that several reviewers legitimately had concerns.

**Note From Pc:**

if the above contains the word "oral" or "spotlight" please see: "oral" presentation means -> notable-top-5% and "spotlight" means -> notable-top-25%. As stated in our emails, we are disassociating presentation type from AC recommendations

**Summary Of Ac-Reviewer Meeting:**

N/A